# Combining rapid diagnostic tests to estimate primary and post-primary dengue immune status at the point of care

**Joseph R. Biggs**[1☯*], **Ava Kristy Sy**[2,3☯], **James Ashall**[1], **Marsha S. Santoso**[4], **Oliver J. Brady**[5,6], **Mary Anne Joy Reyes**[2,3], **Mary Ann Quinones**[2,3], **William Jones-Warner**[1], **Amadou O. Tandoc**[2], **Nemia L. Sucaldito**[7], **Huynh Kim Mai**[8], **Le Thuy Lien**[8], **Hung Do Thai**[8], **Hien Anh Thi Nguyen**[9], **Dang Duc Anh**[9], **Chihiro Iwasaki**[10], **Noriko Kitamura**[10], **Marnix Van Loock**[11], **Guillermo Herrera-Taracena**[12], **Joris Menten**[13], **Freya Rasschaert**[11], **Liesbeth Van Wesenbeeck**[11], **Sri Masyeni**[14], **Sotianingsih Haryanto**[15], **Benediktus Yohan**[4], **Eva Cutiongco-de la Paz**[16,17], **Lay-Myint Yoshida**[10], **Stephane Hue**[5,6], **Maria Rosario Z. Capeding**[3,16], **Carmencita D. Padilla**[16,17], **R. Tedjo Sasmono**[4], **Julius Clemence R. Hafalla**[1], **Martin L. Hibberd**[1,16,17]

1 Department of Infection Biology, Faculty of Infectious and Tropical Diseases, London School of Hygiene and Tropical Medicine, London, United Kingdom, 2 Department of Virology, Research Institute for Tropical Medicine, Manila, Philippines, 3 Dengue Study Group, Research Institute for Tropical Medicine, Manila, Philippines, 4 Dengue Research Unit, Eijkman Institute for Molecular Biology, National Agency for Research and Innovation of the Republic of Indonesia, Jakarta, Indonesia, 5 Department of Infectious Disease Epidemiology, Faculty of Epidemiology and Population Health, London School of Hygiene and Tropical Medicine, London, United Kingdom, 6 Centre for the Mathematical Modelling of Infectious Diseases, London School of Hygiene and Tropical Medicine, London, United Kingdom, 7 Philippine Epidemiology Bureau, Department of Health, Manila, Philippines, 8 Pasteur Institute of Nha Trang, Nha Trang, Vietnam, 9 National Institute of Hygiene and Epidemiology, Hanoi, Vietnam, 10 Paediatric Infectious Diseases Department, Institute of Tropical Medicine, Nagasaki University, Nagasaki, Japan, 11 Janssen Global Public Health, Janssen Pharmaceutica NV, Beerse, Belgium, 12 Janssen Global Public Health, Janssen Research & Development, Horsham, Pennsylvania, United States of America, 13 Quantitative Sciences, Janssen Pharmaceutica NV, Beerse, Belgium, 14 Department of Internal Medicine, Faculty of Medicine and Health Sciences, Universitas Warmadewa, Denpasar, Bali, Indonesia, 15 Raden Mattaher Hospital, Jambi, Indonesia, 16 Institute of Human Genetics, University of the Philippines, Manila, Philippines, 17 Philippine Genome Centre, University of the Philippines, Manila, Philippines

☯ These authors contributed equally to this work.
* Joseph.Biggs1@lshtm.ac.uk

**Data Availability Statement:** All relevant data are included within the manuscript and supporting information files. The data underlying the results

## Abstract

### Background

Characterising dengue virus (DENV) infection history at the point of care is challenging as it relies on intensive laboratory techniques. We investigated how combining different rapid diagnostic tests (RDTs) can be used to accurately determine the primary and post-primary DENV immune status of reporting patients during diagnosis.

### Methods and findings

Serum from cross-sectional surveys of acute suspected dengue patients in Indonesia (N:200) and Vietnam (N: 1,217) were assayed using dengue laboratory assays and RDTs. Using logistic regression modelling, we determined the probability of being DENV NS1, IgM and IgG RDT positive according to corresponding laboratory viremia, IgM and IgG ELISA

presented in the study are available from the LSHTM chief operating officer (coo@lshtm.ac.uk).

**Funding:** This study was funded by grants awarded to M.L.H from Newton Fund's Institutional Links (Project number: 216416089), British Council (BC), the Philippines' Commission on Higher Education (CHED), Janssen (Johnson & Johnson, Project number: ITPMZG4810), PT ASTRA International Tbk, and the Indonesia Ministry of Research, Technology, and Higher Education. O.J.B was supported by a Wellcome Trust Sir Henry Wellcome Fellowship (206471/Z/17/Z) and a UK Medical Research Council Career Development Award (MR/V031112/1). The funders had no role in study design, data collection and analysis, decision to publish, or preparation of the manuscript.

**Competing interests:** I have read the journal's policy and the authors of this manuscript have the following competing interests: MVL, GHT, JM, FR and LVM are employees of Johnson & Johnson.

metrics. Laboratory test thresholds for RDT positivity/negativity were calculated using Youden's J index and were utilized to estimate the RDT outcomes in patients from the Philippines, where only data for viremia, IgM and IgG were available (N:28,326). Lastly, the probabilities of being primary or post-primary according to every outcome using all RDTs, by day of fever, were calculated. Combining NS1, IgM and IgG RDTs captured 94.6% (52/55) and 95.4% (104/109) of laboratory-confirmed primary and post-primary DENV cases, respectively, during the first 5 days of fever. Laboratory test predicted, and actual, RDT outcomes had high agreement (79.5% (159/200)). Among patients from the Philippines, different combinations of estimated RDT outcomes were indicative of post-primary and primary immune status. Overall, IgG RDT positive results were confirmatory of post-primary infections. In contrast, IgG RDT negative results were suggestive of both primary and post-primary infections on days 1–2 of fever, yet were confirmatory of primary infections on days 3–5 of fever.

## Conclusion

We demonstrate how the primary and post-primary DENV immune status of reporting patients can be estimated at the point of care by combining NS1, IgM and IgG RDTs and considering the days since symptoms onset. This framework has the potential to strengthen surveillance operations and dengue prognosis, particularly in low resource settings.

## Author summary

Combined NS1, IgM and IgG dengue rapid diagnostic tests (RDTs) have previously been shown to accurately diagnose those experiencing dengue virus (DENV) infections at the point of care and are now available as single commercial kits. Using such kits to additionally determine those experiencing primary (first) or post-primary (second, third or fourth) dengue infections however remains challenging as accurate immune status classification currently relies on laboratory analysis. We used logistic regression modelling methods to estimate RDT positive and negative outcomes according to corresponding PCR and ELISA laboratory-based methods, which showed high sensitivity and specificity. Dengue RDT outcomes were then predicted among a large sample of suspected dengue case reports, to calculate the probability of being primary or post-primary for dengue according to every possible set of dengue RDT outcomes, by day of fever. Different RDT outcomes, at certain stages of infection, were indicative of primary and post-primary immune status. Using our framework to determine dengue immune status at the point of care in low resource settings, regional surveillance systems could estimate and monitor dengue transmission intensity. Additionally, this framework could potentially support dengue prognosis and identify primary cases who would benefit from current vaccination regimes to prevent subsequent secondary infections associated with severe disease.

## Introduction

Dengue is an emerging arboviral infectious disease, transmitted through the bite of an *Aedes* mosquito that burdens much of the urbanised tropical and subtropical world. According to World Health Organisation (WHO) figures, global case reporting has risen 8-fold in the past

20 years with a record 5.2 million reports in 2019 [1]. However, these data only account for the minority of symptomatic dengue cases who actively sought care and were successfully documented. Therefore, figures exclude most of the 105–390 million annual estimated dengue infections across the globe [2, 3]. Dengue virus (DENV) is a member of the *Flaviviridae* family consisting of four distinct serotypes (DENV1-4) [4]. A primary infection with any serotype is typically associated with a self-limiting disease which elicits a long-lived IgG response that protects against subsequent homologous serotype infections [5]. Consequently, individuals can suffer successive, post-primary (i.e. secondary, tertiary and quaternary), DENV infections during their lifetime with heterologous serotypes. A secondary DENV infection is associated with more severe clinical outcomes, including severe organ impairment and bleeding [6, 7], due to a phenomenon known as antibody-dependent enhancement [8, 9]. Previously elicited, sub-neutralising, IgG resurges upon infection and intensifies viral replication to trigger immuno-modulated severe disease. Without specific dengue chemotherapies, severe disease management is currently limited to intravenous hydration therapy (IHT) that requires careful monitoring and adequate health care infrastructure [10].

Current dengue diagnostics are primarily concerned with capturing active infections, thus no such method for determining primary or post-primary DENV infections, at the point of care, exists [6]. Nonetheless, WHO serological laboratory techniques can be utilized to distinguish DENV immune status retrospectively using patient paired sera collected at the acute and convalescent stage of disease [11]. By assaying for changes in both DENV IgM and IgG antibodies, a rise in IgM titres coupled with high and low convalescent IgM:IgG ratios indicates active primary and secondary infections, respectively. A major caveat to this approach however is the necessity of paired sera which makes it impractical for large scale epidemiological studies and it detects the result too late to inform many case management decisions. Fortunately, more recent studies have shown dengue immune status can be determined using single serum samples collected during the acute phase of disease according to disease day-specific IgG:IgM ratios [12–15]. Yet again however, these algorithms have limited value in point-of-care testing, particularly in low resource settings, as they rely on individual laboratory metrics which take time to generate, are labour intensive and require extensive equipment.

Recently, the WHO has advocated for the use of rapid diagnostic tests (RDTs) to improve dengue case detection and management in low resource, regional health care facilities [16]. Numerous quick, easy-to-use and inexpensive commercial RDTs are now available which can detect different markers of infection [17, 18]. The dengue NS1 RDT detects the dengue non-structural protein 1 (NS1), which is secreted into the blood during, and slightly after, the viraemic phase of disease. In contrast, The IgM and IgG RDTs detect IgM and IgG antibodies during the later immunogenic phase of infection, respectively. Many studies have shown how well these kits capture true active dengue infections, particularly when used in combination, although their performance varies according to specific commercial brands [17, 19–21]. Moreover, studies have documented how the DENV immune status of patients influences the performance of RDTs. When used alone, NS1 RDTs have better sensitivities for capturing primary, as opposed to secondary, infections [18, 22]. This is likely a consequence of the contrasting viremia kinetics, whereby viremia is higher and persists longer in the febrile stage during primary compared to post-primary infections [12, 23, 24]. Conversely, studies have demonstrated IgG RDTs are better at capturing post-primary infections as IgG is believed to be absent during the acute phase of primary disease [22, 25]. Although, it has been shown during a primary infection, patients can begin to elicit IgG towards the end of the acute phase while very early stage post-primary infections still experience increasing titres of IgG which may not be high enough to yield IgG RDT positive results [12, 14]. Indeed one study revealed the IgG RDT sensitivity for capturing secondary DENV infections was lower among those reporting before disease day 4 [26]. Consequently, assuming all primary and post-

primary dengue infections would yield IgG RDT negative and positive results, respectively, could result in misclassification. Despite studies demonstrating that combining the three NS1, IgM and IgG RDTs improves diagnostic performance, it remains unclear what exact combination of RDT outcomes, at specific stages of infection, indicate primary or post-primary dengue infections.

Documenting DENV immune status at the point of care could assist surveillance operations. The age at which patients present with their first (primary) DENV infection has been shown to correlate with the force of infection in endemic cities [27] and establishing immune status promptly might assist in the deployment of vaccinations targeted at those with primary DENV exposure [28]. Moreover, as patients can deteriorate quickly during a DENV infection, determining primary and post-primary immune status prior to the development of severe symptoms could potentially assist in clinical case management in health care settings [1]. For instance, post-primary DENV patients could receive closer monitoring and be prioritised for limited IHT compared to primary DENV patients. Yet whether this would be appropriate for effective case management remains unknown and would require further investigation.

Prior to this study, we generated and validated an algorithm capable of distinguishing individual primary and post-primary DENV immune status that relies on basic epidemiological and laboratory-obtained metrics from single serum samples [12]. The framework utilises individual molecular and serological DENV metrics from the CDC fourplex DENV1-4 PCR assay and commercial IgM and IgG capture ELISAs (Panbio, Abbott, Cat no. 01PE10 & 01PE20), respectively. Panbio serological assays were chosen based on a WHO report which compared their performance to other commercial assays which revealed similar sensitivities [29]. The novel algorithm achieved 90% agreement with the WHO gold standard method for categorising immune status based on paired sera and proved superior to the Panbio method for classifying immune status [11, 12]. Given dengue serological assays have been shown to detect cross-reactive antibodies elicited from other flavivirus infections, including Zika virus (ZIKV) which is often co-endemic with dengue [30–32], previous analysis explored the impact of ZIKV exposure on the generated DENV immune status algorithm [12]. Results revealed a proportion of those with post-primary, opposed to primary, DENV infections had evidence of ZIKV IgG but not IgM exposure. This suggested some patients had historical ZIKV exposure that primed individuals for a subsequent post-primary-like, instead of a primary-like, DENV infection upon their first infection with DENV. Moreover, further analysis suggested post-primary DENV infections with prior ZIKV exposure were at risk of adverse clinical symptoms [32] which has been previously reported in Nicaragua [33]. These findings suggested that individuals categorised as post-primary DENV infections include those with either prior exposure to DENV and/or other flaviviruses [12]. A major challenge associated with the generated immune status algorithm is the reliance on laboratory-derived metrics (PCR and ELISA), consequently this framework has limited value in low-resource, regional health care settings [16]. In this study, we investigated whether RDTs can be utilised to accurately determine primary and post-primary immune status of reporting patients at the point of care. Specifically, we examined: 1) the utility of combining the outcomes of NS1, IgM and IgG RDTs in accurately capturing both primary and post-primary dengue infections; 2) the translatability between dengue laboratory tests and RDTs; and 3) The probability of being primary or post-primary dengue cases by every possible NS1, IgM and IgG RDT outcome at specific days of disease.

## Methods

### Ethics statement

This study was approved by the ethical review boards of the London School of Hygiene and Tropical Medicine (Ref: 17853), the Research Institute for Tropical Medicine (Ref: 2017–014),

Nagasaki University/The National Institute of Hygiene and Epidemiology, Vietnam (Ref: VN01057) and Eijkman Institute for Molecular Biology (Ref: 136/2019). Verbal consent was obtained from patients over 18 years, while verbal assent was acquired from those under 18 years coupled with parent/guardian consent, for the use of serum samples. All unique participant identifiers were removed before data acquisition.

## Data collection

Data were obtained from suspected dengue patients who visited health care facilities during the acute stage of disease in the Philippines (N: 28,326), Vietnam (N: 1,217) and Indonesia (N: 200). Suspected dengue patients included those with a self-reported sudden acute fever coupled with at least two additional warning signs: headache, malaise, myalgia, arthralgia, retro-orbital pain, anorexia, nausea, vomiting, diarrhoea, flushed skin and/or rash in accordance with WHO criteria [6]. Specific data collected from patients in each dataset are highlighted in (S1 Table).

In the Philippines, data were collected from a survey of dengue patients who visited disease reporting units (DRUs) situated across the country between 2014 and 2018. In major, regional DRUs, five weekly random serum samples were collected from suspected dengue patients during the acute phase of disease. In smaller regional health care centres across the Philippines, samples were collected from patients during an upsurge in case reporting defined according to Philippine Integrated Disease Surveillance and Response (PIDSR) criteria [34]. Additional epidemiological data were collected from patients including age, sex, disease day (date of reporting–date of symptom onset), symptoms (no warning signs, warning signs, severe dengue). Serum samples were sent to the Research Institute of Tropical Medicine (Department of Health, Manila, Philippines) for further laboratory testing.

In Vietnam, data used in this study were obtained from those who reported with suspected dengue to a Polyclinic or the Tropical Disease Hospital out-patient clinic in Nha Trang city between October 2016 and May 2019. We enrolled patients who gave home addresses from four communes in Nha Trang City: Vinh Hai, Vinh Phuoc, Vinh Tho, and Vinh Hoa. Serum samples underwent subsequent laboratory testing at the Pasteur Institute in Nha Trang. Epidemiological data collected from patients included: Age, sex, symptoms, and disease day.

In Indonesia, serum samples were collected from suspected dengue patients that reported across regions of Indonesia between July 2014 and July 2019 originally obtained for a previous study [18]. Additional epidemiological data provided for each sample included age and disease day. Samples were stored and assayed at the Eijkman Institute, Jakarta, Indonesia.

## Dengue testing

All serum samples collected from patients included in this study (N: 29,743) received laboratory dengue testing in their respective institutes. Samples were assayed for the presence of DENV1-4 viremia using the CDC fourplex, real-time polymerase chain reaction (RT-PCR) test according to methods described in [35]. Briefly, dengue serotype-specific primers amplify viral RNA and yield critical threshold (Ct) values which inversely corresponds to the level of viral RNA (viremia). Samples with Ct values $\leq 36$ were considered PCR positive for DENV. The presence of DENV IgM and IgG antibodies was performed using Panbio capture ELISAs according to manufacturer's instructions (Cat no: 01PE20; 01PE21, Abbott). Assays detect IgM/G antibodies specific to all serotypes and provide plate-calibrated titre outputs termed 'panbio units'.

Additional laboratory and RDT testing were conducted among samples obtained from Vietnam and Indonesia. In Vietnam, patients were tested, at the point of care in the Polyclinic,

for the presence and absence of NS1 using DENV NS1 RDTs (Cat no: 70700, Bio-Rad, Inc) (N: 1,217). Among samples from Indonesia, patient serum samples were tested for DENV NS1 using both NS1 capture ELISAs (Cat no: 01PE40, Abbott) and NS1 RDTs (Cat no: 09DEN10D, SD Biosensor) according to commercial guidelines and in the laboratory. NS1 capture ELISAs generated plate-calibrated titres termed 'NS1 panbio units'. Finally, samples collected from Indonesia were further tested for the presence or absence of DENV IgM/G using IgM and IgG RDTs in line with manufacturers specifications in the laboratory (Cat no: 09DEN20D, SD Biosensor). A summary of the data collected from reporting patients in each country are shown in (S1 Table).

## Statistical analysis

Using laboratory and basic epidemiological data, we categorized the reference DENV immune status (primary, post-primary, historical and negative) of the entire study population using the exact methods described in [12]. Patients who reported as either PCR+ or IgM+ (Ct≤36 or IgM panbio units>9.9) were classified as active DENV infections as both these markers are detectable during infection. PCR- and IgM- (IgM panbio units<9.9) cases were categorised as non-active DENV infections and represent patients misdiagnosed as suspected active dengue. Non-active DENV infections were further classified as historical or negative if they were DENV IgG positive (IgG panbio units>2.2) or negative (IgG panbio units<2.2), respectively. Historical and negative cases included misdiagnosed patients who reported without a current dengue infection yet with and without previous exposure to DENV, respectively. IgG:IgM ratios (IgG panbio units/IgM panbio units) were used to distinguish active DENV infections as primary or post-primary cases. Among active dengue patients at the early stage of disease (disease day 1 or 2), those DENV IgG+ and IgG- were classed as post-primary and primary respectively. Among active cases on disease day 3 to 5, individuals with IgG:IgM ratios above and below 0.45 were categorised as post-primary and primary respectively. As a consequence of previous findings [12, 32], post-primary cases included current DENV infections with at least one previous flavivirus infection including DENV and or ZIKV. An overview of the reference DENV immune status classification is shown in S1 File.

Using binomial logistic regression modelling, we estimated the probability of being RDT positive according to corresponding laboratory-derived metrics with 95% confidence intervals (ELISA & PCR). Using data from Indonesia, we estimated the probability of being IgM and IgG RDT positive according to IgM and IgG panbio units, respectively. From the Vietnam dataset, we predicted the probability of being NS1 RDT positive according to DENV viremia (Ct value). To account for the lag in NS1 production during the viraemic stage of infection, we stratified NS1 logistic regression models by disease day. To assess the validity of logistic regression modelling, Hosmer–Lemeshow tests were used to determine appropriate model fits (p-value>0.05). For each model, the optimal laboratory-derived metric cut off for RDT positivity was determined using Youden's J index (sensitivity + specificity– 1) [36]. The threshold refers to the optimal estimated probability of being RDT positive according to sensitivity/specificity based on actual RDT outcomes. This approach was adopted to minimise the misclassification of RDT outcomes according to corresponding laboratory metrics. Moreover, the percentage agreement between of the combined (NS1, IgM and IgG) estimated and actual RDT outcomes were calculated. To estimate immune status according to RDTs, we estimated the NS1, IgM and IgG RDT status of all patients from the Philippines with defined primary, post-primary, historical and negative DENV immune status according to laboratory testing (S1 File). Lastly, we calculated the probability of being primary, post-primary, historical and negative according

to every combination of RDT result possible using all three rapid tests, stratified by disease day.

## Results

In our study population, we observed similar demographic characteristics among suspected dengue patients who reported in Indonesia, Vietnam, and the Philippines (S2 Table). Most were aged between 6–15 years ($\geq$33.4%), reported 3–4 days after the onset of disease symptoms ($\geq$51.0%) and presented with post-primary DENV infections ($\geq$48.7%). There were contrasting patterns in the DENV serotypes patients presented with. In Vietnam only 0.5% (4/803) of patients assayed by PCR were DENV-3, while among those assayed for PCR in the Philippine dataset, 17.1% were DENV-3 (4535/26,494).

### Dengue infection kinetics

We explored disease-day stratified DENV infection kinetics among primary and post-primary dengue patients according to the laboratory and RDT data collected among patients from Indonesia (Fig 1) (N:200). Viremia, as measured by Ct value, and NS1 levels, measured by ELISA, plateaued at higher levels during the acute phase of primary infections (Fig 1A and 1B) yet were lower and dropped more rapidly during the acute phase of post-primary infections (Fig 1E and 1F). This was mirrored by the higher proportion of primary cases who were NS1 RDT positive (81.8% 45/55) compared to post-primary cases (45.9% 50/109) during the acute phase of disease (Fig 1I and 1M). Likewise, we found both IgM and IgG RDT outcomes matched IgM and IgG ELISA laboratory values, respectively. For IgG, ELISA titres among primary cases remained low during the acute phase with only 20.0% (11/55) IgG RDT positive (Fig 1D and 1K). In contrast, median IgG ELISA values increased to high levels among post-primary cases (Fig 1H) which was reflected by an increase in IgG RDT positivity from 13.3% (2/15) to 78.4% (29/37) on disease days 1–2 and 4–5, respectively (Fig 1O). Given not all early-stage post-primary cases yielded IgG RDT positive outcomes, these results indicate assuming post-primary and primary DENV cases would present as IgG RDT positive and negative, respectively, would result in immune status misclassification. Lastly, we found combining RDTs maximised the chances of identifying primary and post-primary DENV infections at all stages of acute disease. For primary and post-primary cases, 94.6% (52/55) and 95.4% (104/109) were positive to either NS1, IgM or IgG RDTs, respectively (Fig 1L and 1P).

Among patients from the Philippines with serotype and PCR (Ct values) data, we explored whether the contrasting viremia kinetics among primary and post-primary cases during the acute stage of disease were driven by the infecting serotype (S1 Fig). Irrespective of serotype, viremia decreased from higher levels in primary cases compared to post-primary cases. We also investigated whether contrasting NS1 patterns observed between primary and post-primary cases was influenced by certain serotypes secreting more/less NS1 proteins (S3 Table). We found NS1 RDTs effectively captured most DENV1-4 PCR+ infections, regardless of immune status (Sensitivity>80% for each serotype).

### Estimating RDT outcomes according to laboratory metrics

To investigate the translatability between laboratory and rapid dengue diagnostics, we explored whether DENV RDT outcomes could be predicted from DENV laboratory test results using logistic regression models. Models were assessed using Hosmer–Lemeshow tests which revealed no significant poor model fits (p-value>0.05) (S2 Fig). Among dengue patients from Indonesia (N:200), logistic regression models were used to estimate the probability of being IgM/G RDT positive according to IgM/G ELISA panbio units, respectively (Fig 2A and 2B).

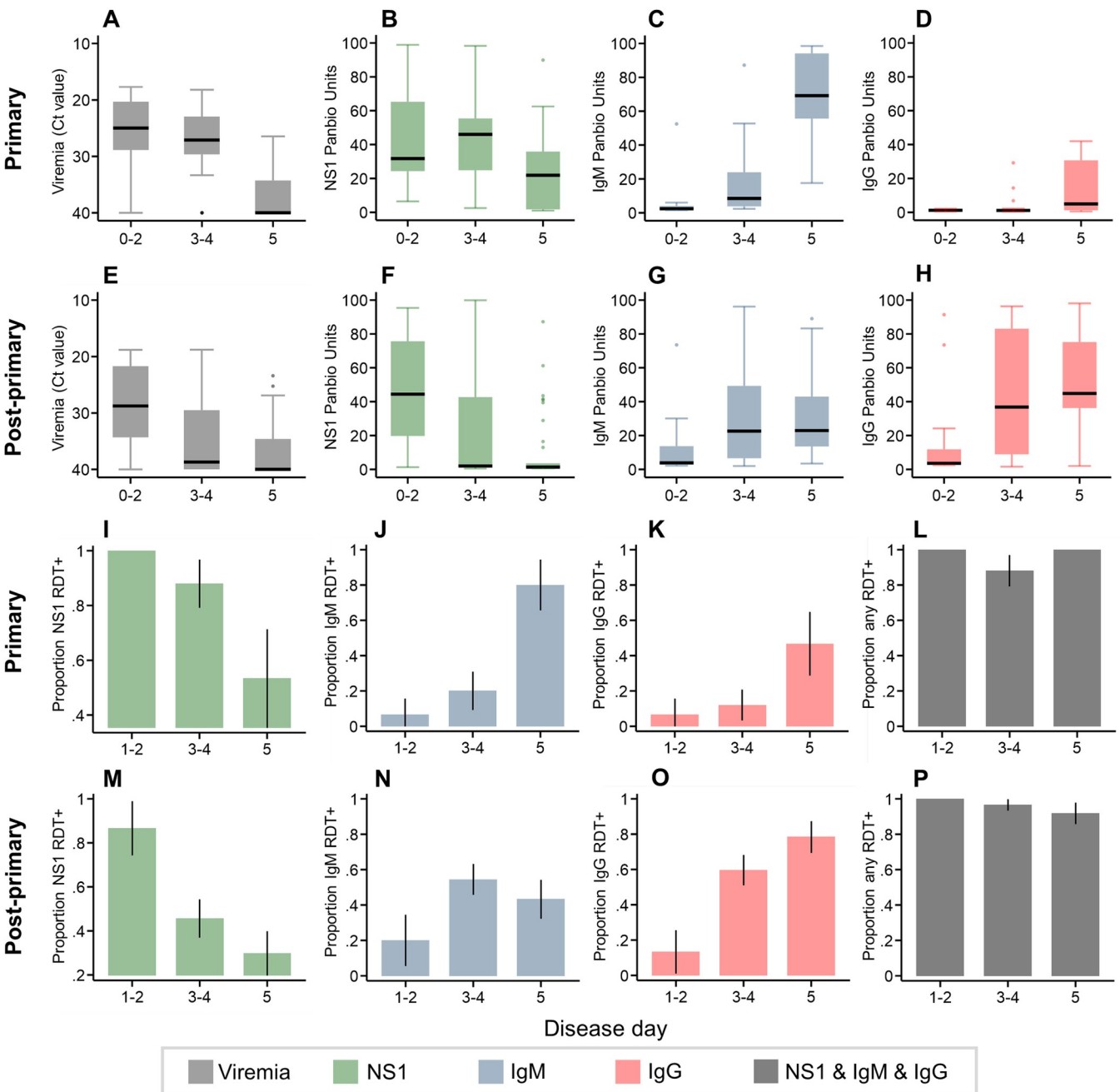

**Fig 1. Infection kinetics among primary and post-primary DENV patients from Indonesia by disease day according to laboratory and rapid tests. A-H:** Boxplots of viremia (Ct), NS1, IgM and IgG by disease day among primary and post-primary cases according to PCR and ELISA tests. **I-P:** Proportion RDT positive to NS1, IgM, IgG and all combined by disease day among primary and post-primary cases. Black error bars: 90% confidence intervals based on t-distributions. (Primary N: 55) (Post-primary N:109).

For both IgM and IgG, the estimated probability of being RDT positive increased with increasing ELISA panbio units. According to the highest Youden's J-index values, thresholds of 15.2 and 11.9 panbio units provided the optimal sensitivity and specificity for determining IgM and IgG RDT positivity, respectively. Individuals with ELISA values below and above these

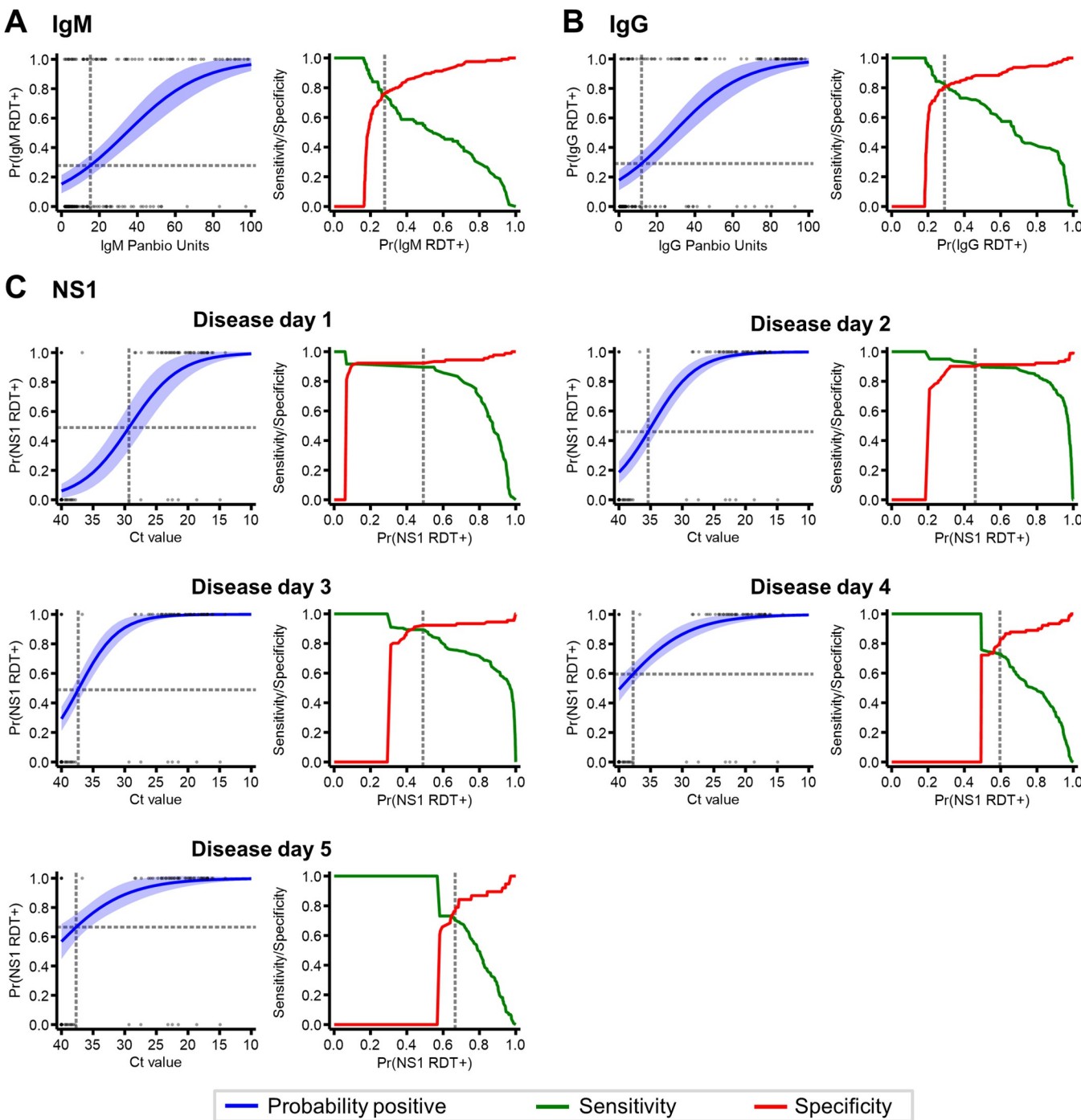

**Fig 2. Estimated probability of being DENV RDT positive according to corresponding laboratory-derived metrics using logistic regression modelling.** A: IgM RDT positivity according to IgM panbio units and the corresponding sensitivity/specificity among patients from Indonesia (N:200) B: IgG RDT positivity according to IgG panbio units and the corresponding sensitivity/specificity among patients from Indonesia (N:200) C: NS1 RDT positivity according to viremia (Ct value) and the corresponding sensitivity/specificity stratified by disease day among patients from Vietnam (N: 1,217). Grey dash: estimated laboratory-derived metric threshold for RDT positivity according to the optimal Youden's J index value.

thresholds were considered RDT positive for each antibody. This resulted in 43.0% (86/200) with estimated IgM RDT positive outcomes which corresponded to 37.5% (75/200) with actual IgM RDT positivity (Sensitivity: 74.7% [95%CI: 63.3–84.0%], Specificity: 76.0% [95%CI: 67.5–

**Table 1. Estimated laboratory-test values that yield RDT positive results compared to actual RDT outcomes.** Estimated RDT positivity threshold refer to the optimal Youden's J index value. Estimated/actual NS1 RDT positivity determined among patients from Vietnam (N: 1,217). Estimated/actual IgM/IgG RDT positivity determined among patients from Indonesia (N:200).

| DENV metric | Disease day | Lab test thresholdᵃ | Estimated RDT positive | | | Actual RDT positive | | | Sensitivity | | Specificity | |
|---|---|---|---|---|---|---|---|---|---|---|---|---|
| | | | % | N | n | % | N | n | % | [95% CI] | % | [95% CI] |
| NS1 | 1 | <29.3 Ct | 36.2 | 138 | 50 | 34.8 | 138 | 48 | 89.6 | [77.3-96.5] | 92.2 | [84.6-96.8] |
| | 2 | <35.4 Ct | 65.9 | 293 | 193 | 68.9 | 293 | 202 | 91.1 | [86.3-94.6] | 90.1 | [82.1-95.4] |
| | 3 | <37.4 Ct | 62.7 | 279 | 175 | 67.4 | 279 | 188 | 89.4 | [84.0-93.4] | 92.3 | [84.8-96.9] |
| | 4 | <37.8 Ct | 56.1 | 289 | 162 | 68.9 | 289 | 199 | 72.9 | [66.1-78.9] | 81.1 | [71.5-88.6] |
| | 5 | <37.7 Ct | 57.8 | 142 | 82 | 73.2 | 142 | 104 | 70.2 | [60.4-78.8] | 76.3 | [59.8-88.6] |
| IgM | - | >15.2 PU | 43.0 | 200 | 86 | 37.5 | 200 | 75 | 74.7 | [63.3-84.0] | 76 | [67.5-83.2] |
| IgG | - | >11.9 PU | 47.0 | 200 | 94 | 44.5 | 200 | 89 | 82.0 | [72.5-89.4] | 81.1 | [72.5-87.9] |

ᵃ Estimated laboratory test value that corresponds to RDT positivity

Ct, Critical threshold; PU, Panbio units

83.2%]) (Table 1). For IgG, our optimal threshold resulted in 47.0% (94/200) with estimated IgG RDT positivity which corresponded to 44.5% (89/200) actual IgG RDT positivity (Sensitivity: 82.0% [95%CI: 72.5–89.4%], Specificity: 81.1% [72.5–87.9%]) (Table 1).

Among patients from Vietnam (N: 1,217), we estimated NS1 RDT outcomes according to DENV viremia (PCR Ct value) (Fig 2C). As PCR assays detect DENV RNA directly from the virus and NS1 RDTs detect virus-secreted proteins that peak during and after viremia, we opted to stratify logistic regression models by disease day to account for the delayed NS1 production. For each day of disease, the probability of being NS1 RDT positive increased with decreasing Ct values (increasing viremia). According to the logistic function however, as day of disease progressed, individuals were more likely to be NS1 RDT positive at lower levels of viremia. For instance, among those with a DENV PCR Ct value of 34, we estimated 22.9% [95%CI: 11.8–33.7%] were NS1 RDT+ on disease day 1 while we estimated 79.6% [95%CI: 71.4–87.6%] were NS1 RDT positive on disease day 5 (Fig 2C). This infers NS1 levels are impacted by both the amount of virus and the stage of infection. Consequently, this yielded disease-day specific NS1 RDT thresholds according to PCR Ct values which increased with disease day (Table 1). Upon predicting NS1 RDT outcomes according to disease day-stratified thresholds, we estimated 36.2% (50/138) were NS1 RDT positive on disease day 1 which corresponded to 34.8% with actual NS1 RDT positive results on disease day 1 (Sensitivity: 89.6% [95%CI: 77.3–96.5%; Specificity: 92.2% [95%CI: 84.6–96.8%]). By disease day 5, this agreement decreased slightly as 57.8% (82/142) and 73.2% (104/142) had estimated and actual NS1 RDT results, respectively (Sensitivity: 70.2% [95%CI: 60.4–78.8%; Specificity: 76.3% [95%CI: 59.8–88.6%]) (Table 1).

After generating DENV PCR and ELISA test thresholds that we estimated gave rise to NS1 and IgM/G RDT positive results, respectively, we explored how well our laboratory thresholds could estimate all 3 RDTs combined. Among the Indonesian sample population (N:200) who were tested using all 3 RDTs, we investigated the combined estimated RDT outcome agreement with the actual combined DENV RDT results (Table 2). Overall, our combined RDT outcome estimates achieved 79.5% (159/200) agreement overall. After stratifying by immune status, estimated and actual RDT agreement for primary and post-primary cases equated to 87.3% (48/55) and 78.0% (85/109), respectively. Together these results demonstrated that we were able to accurately determine the outcomes of DENV RDTs according to patient DENV laboratory metrics.

**Table 2. Agreement between the estimated and actual combined DENV RDT results of patients in Indonesia.**

| DENV infection status | | Combined RDT agreement | | |
|---|---|---|---|---|
| | | % | n | N |
| **Age** | | | | |
| | 0-5 | 82.4 | 28 | 34 |
| | 6-15 | 78.2 | 68 | 87 |
| | 16-30 | 82.6 | 38 | 46 |
| | $\geq$31 | 75.8 | 25 | 33 |
| **Disease day** | | | | |
| | 1-2 | 84.2 | 32 | 38 |
| | 3-4 | 79.4 | 81 | 102 |
| | 5 | 76.7 | 46 | 60 |
| **Serotype** | | | | |
| | DENV-1 | 92.0 | 23 | 25 |
| | DENV-2 | 80.0 | 20 | 25 |
| | DENV-3 | 80.0 | 20 | 25 |
| | DENV-4 | 76.0 | 19 | 25 |
| | PCR- | 77.0 | 77 | 100 |
| **DENV immune status** | | | | |
| | Primary | 87.3 | 48 | 55 |
| | Post-primary | 78.0 | 85 | 109 |
| | Historical | 70.0 | 14 | 20 |
| | Negative | 75.0 | 12 | 16 |
| **Total** | | 79.5 | 159 | 200 |

## Combining RDTs to estimate primary and post-primary DENV status

According to the optimal dengue laboratory metric thresholds, we estimated the NS1, IgM and IgG RDT positive and negative status of study population in the Philippines which lacked

**Table 3. The probability of being primary, post-primary or historical for DENV according to every outcome combination of NS1, IgM and IgG RDTs stratified by disease day. RDT results** estimated among patients from across the Philippines (N: 28,326).

| Estimated RDT result | Disease day 1 | | | | 2 | | | | 3 | | | | 4 | | | | 5 | | | |
|---|---|---|---|---|---|---|---|---|---|---|---|---|---|---|---|---|---|---|---|---|
| | Total | Probability | | | Total | Probability | | | Total | Probability | | | Total | Probability | | | Total | Probability | | |
| | | 1° | 2° | Hist | | 1° | 2° | Hist | | 1° | 2° | Hist | | 1° | 2° | Hist | | 1° | 2° | Hist |
| **1 positive RDT** | | | | | | | | | | | | | | | | | | | | |
| NS1+ IgM- IgG- | 248 | 0.51 | 0.49 | 0.0 | 460 | 0.47 | 0.53 | 0.0 | 612 | 0.36 | 0.64 | 0.0 | 299 | 0.44 | 0.56 | 0.0 | 78 | 0.46 | 0.54 | 0.0 |
| NS1- IgM+ IgG- | 180 | 0.45 | 0.55 | 0.0 | 224 | 0.51 | 0.49 | 0.0 | 364 | 0.98 | 0.02 | 0.0 | 376 | 0.99 | 0.01 | 0.0 | 270 | 0.99 | 0.01 | 0.0 |
| NS1- IgM- IgG+ | 160 | 0.0 | 0.48 | 0.52 | 246 | 0.0 | 0.44 | 0.56 | 388 | 0.0 | 0.46 | 0.54 | 308 | 0.0 | 0.49 | 0.51 | 184 | 0.0 | 0.47 | 0.53 |
| **2 positive RDTs** | | | | | | | | | | | | | | | | | | | | |
| NS1+ IgM+ IgG- | 18 | 0.33 | 0.67 | 0.0 | 118 | 0.48 | 0.52 | 0.0 | 237 | 0.97 | 0.03 | 0.0 | 265 | 0.97 | 0.03 | 0.0 | 98 | 0.96 | 0.04 | 0.00 |
| NS1+ IgM- IgG+ | 19 | 0.0 | 1.0 | 0.0 | 119 | 0.0 | 1.0 | 0.0 | 199 | 0.0 | 1.0 | 0.0 | 151 | 0.0 | 1.0 | 0.0 | 54 | 0.0 | 1.0 | 0.0 |
| NS1- IgM+ IgG+ | 551 | 0.0 | 1.0 | 0.0 | 908 | 0.0 | 1.0 | 0.0 | 1613 | 0.0 | 1.0 | 0.0 | 1747 | 0.0 | 1.0 | 0.0 | 926 | 0.0 | 1.0 | 0.0 |
| **3 positive RDTs** | | | | | | | | | | | | | | | | | | | | |
| NS1+ IgM+ IgG+ | 6 | 0.0 | 1.0 | 0.0 | 204 | 0.0 | 1.0 | 0.0 | 658 | 0.0 | 1.0 | 0.0 | 818 | 0.0 | 1.0 | 0.0 | 331 | 0.0 | 1.0 | 0.0 |

1°, Primary DENV; 2° Post-primary DENV; Hist, Historical DENV infection

RDT data (N: 28,326). For every possible RDT outcome using all three tests by disease day, we calculated the probability of being primary, post-primary and historical for dengue (Table 3). It should be noted, all those with at least one predicted positive RDT result were either primary, post-primary or historical for dengue. The most common combination of RDT outcomes in the study population was NS1-, IgM+ and IgG+ (5,745) while the least common was NS1+, IgM- and IgG+ (542). For many combinations of RDT outcomes on specified disease days, RDT results corresponded to very clear immune status outcomes. The presence of an IgG+ RDT result nearly always represented a post-primary DENV infection. For instance, on disease day 3, 100% (1,613/1,613) of patients with an estimated NS1- IgM+ IgG+ RDT outcome combination were post-primary dengue infections. At the early stages of infection (disease day 1–2), IgG negative RDT results yielded uninformative immune status outcomes. Yet towards the later stages of acute disease (disease day 3–5), IgG negative RDT results were often confirmatory of primary infections. For instance, patients with estimated NS1- IgM+ IgG- RDT outcomes on disease days 4 and 5 had a >99% probability of being a primary case. These results reveal certain combinations of RDT results, at different stages of infection, can be confidently used to determine immune status while some combinations yield more uncertain conclusions.

## Discussion

In this study, we demonstrated that dengue rapid tests corresponded well to associated laboratory metrics and that combining different types of RDTs accurately captured laboratory-determined primary and post-primary DENV infections. At certain stages of an acute DENV infection, different combination of NS1, IgM and IgG RDT results gave rise to clear predictions of immune status, yet at other stages of disease, ambiguous immune status classifications were estimated. We found that IgG RDT positivity was almost always confirmatory of a post-primary DENV infection. In contrast, an IgG RDT negative result on fever days 1 and 2 were suggestive of both primary and post-primary infections while at fever days 3 to 5 were confirmatory of a primary infection. This infers simply classifying reporting primary and post-primary DENV cases according to IgG RDT negative and positive results, respectively, would lead to immune status misclassification.

As shown previously, combining NS1, IgM and IgG DENV RDTs maximises the chances of capturing both primary and post-primary dengue infections and that using NS1 RDTs individually, risks misdiagnosing DENV infections [17, 22]. We revealed the poor performance of NS1 RDTs in diagnosing post-primary cases is attributed to the lower overall viremia post-primary cases experience during the acute phase of disease (relative to primary cases)—a trend that has been shown before [23, 24]. It has been suggested that enhanced, T-cell modulated, viral clearance may account for patients with post-primary dengue to present with lower viremia than primary cases [37, 38]. Alternatively, post-primary cases could just be typically reporting earlier than primary infections [39]. Concerning IgG RDTs, we found many early acute stage (fever day 1–2) post-primary infections were IgG RDT negative due to their low IgG titres. This may be a consequence of pre-elicited IgG titres rising from low levels during the early stage of a post-primary infection which are not high enough to generate a positive IgG RDT result due to elevated test thresholds [25].

In our study, we revealed individual laboratory metrics (PCR, IgM, IgG) were good predictors of corresponding NS1 IgM and IgG RDT outcomes. Despite this, we did observe some discordance between RDT results and laboratory metrics. For instance, several individuals with low antibody ELISA values still produced IgM/G RDT positive results. This might be due to the contrasting commercial brands used for the ELISAs and RDTs that rely on different

epitopes present on DENV antigen that have contrasting immunogenicities. In contrast, some with elevated ELISA antibody response were negative for corresponding antibody RDTs. This trend could be attributed to ELISAs, yet not RDTs, cross-reacting with other flaviviruses including ZIKV which has shown to be potentially widespread across dengue-endemic countries [30–32]. It is now well established commercial DENV ELISAs cross-react with ZIKV [40, 41] yet whether DENV antibody rapid tests cross-react with ZIKV remains poorly characterised and deserves further attention [25, 42]. For NS1, we found as the disease progressed, the probability of being RDT positive increased for any level of viremia. This is likely a consequence of the time lag between DENV viremia and NS1 secretion whereby NS1 proteins persist longer in the bloodstream than detectable nucleic acid [23, 43]. This was likely a key factor for why we were less able to accurately predict NS1 RDT positivity later during the acute phase of disease. By disease day 5, our models predicted Ct value of less than 37 (very low viremia) had >50% probability of yielding a NS1 RDT positive result. Overall however, our estimated combined RDT outcomes achieved a high level of agreement with actual RDT outcomes in the study population demonstrating we could reasonably estimate the RDT status of those without RDT data.

Our combined rapid test framework for determining primary and post-primary dengue immune status has the potential to assist dengue control efforts. It could strengthen regional surveillance systems in settings where laboratory testing is unfeasible [16]. For instance, health care workers could utilise the framework to calculate the age of those reporting primary infections to estimate and monitor the dengue force of infection as described in [27]. Furthermore, this framework could be used to inform vaccination deployment. Currently, the only fully licensed vaccine against dengue, Dengvaxia, is recommended to those with prior dengue exposure in endemic areas aged between 9 and 45 years [44]. This is to ensure dengue-naïve recipients are not primed for a subsequent severe secondary infection by vaccination [45, 46]. However, current screening methods are unable to distinguish those with one or multiple previous infection(s) [47]. Consequently, numerous individuals could be targeted, at cost, for vaccination yet would not benefit from the protection as they may have experienced multiple dengue infections beforehand. Our framework could be used to identify reporting patients with primary infections who represent suitable targets for vaccination. Moreover, monitoring the age of reporting primary infections in certain settings could be informative for population-based pre-vaccination screening. If in high endemicity areas patients report with their first dengue infection at an earlier age than 9 years, this could warrant other younger children in these areas for pre-vaccination screening.

Our rapid test framework also has the potential to benefit dengue case management. Given a secondary DENV infection is a risk factor for severe disease [5, 48], determining immune status using these simple point-of-care tests could assist health care workers in prioritising patients for further monitoring and additional supportive treatment [10]. However, it should be noted that most suspected dengue cases who report to health facilities are post-primary DENV infections as these infections are associated with more symptomatic outcomes than primary infections [49, 50]. Consequently, prioritising all post-primary patients for additional severe disease monitoring in health care facilities would likely be unviable. Nonetheless, there are other potential prognostic markers of severe disease, including serum chymase [51] NS1 [52] and RNA/proteins [53]. Furthermore, it has been previously shown that post-primary dengue infections under the age of 10 years are at greater risk of severe disease than those over ten years [27]. Therefore, whether this immune status rapid test framework could be integrated with other prognostic markers into an early severe disease warning system, such as those described in [54, 55], warrants further investigation.

There are some noteworthy limitations associated with this study. Firstly, our results are limited to the commercial diagnostics used in this analysis. Other commercial kits may have varying sensitivities and specificities that may yield slightly contrasting results. Despite this, our work provides a methodological framework for other kits to be evaluated. Secondly, the accuracy of this immune status RDT framework was based on a laboratory immune status framework that had 90.5% serological agreement with the gold standard WHO method for categorising primary and secondary dengue [12]. Therefore, our accuracy estimates are likely slightly overestimated. Lastly, our combined RDT outcome classification of immune status is based upon estimated, not actual, RDT results. This was necessary as just estimating the immune status based on the minority with actual RDT results would yield less confident results.

## Conclusions

We describe methods for estimating the primary and post-primary immune status of dengue patients at the point of care, using a combination of simple-to-use rapid diagnostic tests. Using all three NS1, IgM and IgG RDTs, we demonstrate how at certain stages of infection health care workers and surveillance operations could confidently determine types of dengue infections. It is hoped our framework might lead to improved dengue case management and disease surveillance by identifying those who may benefit from close monitoring and could be utilised to estimate dengue transmission intensity.

## Supporting information

**S1 Table. A summary of data collected from suspected dengue patients included in the study population from Indonesia, Vietnam and the Philippines.**
(XLSX)

**S2 Table. Demographic characteristics of the reporting study population at enrolment.**
Includes suspected dengue case reports who reported in Indonesia (N:200), Vietnam (N:1,217) and the Philippines (N: 28,326).
(XLSX)

**S3 Table. Sensitivity (95%CI) of NS1 RDTs in capturing DENV1-4 infections stratified by immune status among patients from Vietnam (N:1,217).**
(XLSX)

**S1 File. An overview of the methods used to characterise the DENV primary and post-primary immune status according to laboratory test methods.**
(DOCX)

**S1 Fig. Disease day stratified patterns in viremia (Ct value) among primary and post-primary dengue cases from across the Philippines (N: 28,326).**
(TIF)

**S2 Fig. Hosmer–Lemeshow plots used to assess the logistic regression model fits for IgM, IgG and NS1 RDTs.** p-values >0.05 infer good model fit.
(DOCX)

## Acknowledgments

We would like to thank those who participated in the study and the health care personnel who collected data from patients across the Philippines, Indonesia, and Vietnam. We thank the

participating staff at the Research Institute for Tropical Medicine, Philippine Epidemiology Bureau, Eijkman Institute for Molecular Biology, Pasteur Institute of Nha Trang and the Hanoi National Institute of Hygiene and Epidemiology for their work in collecting and managing data.

## Author Contributions

**Conceptualization:** Joseph R. Biggs, Ava Kristy Sy, Eva Cutiongco-de la Paz, Maria Rosario Z. Capeding, Carmencita D. Padilla, R. Tedjo Sasmono, Julius Clemence R. Hafalla, Martin L. Hibberd.

**Data curation:** Joseph R. Biggs, Ava Kristy Sy, James Ashall, Marsha S. Santoso, Nemia L. Sucaldito, Hien Anh Thi Nguyen, Dang Duc Anh, Sri Masyeni, Sotianingsih Haryanto.

**Formal analysis:** Joseph R. Biggs.

**Funding acquisition:** Ava Kristy Sy, Marnix Van Loock, Guillermo Herrera-Taracena, Joris Menten, Freya Rasschaert, Liesbeth Van Wesenbeeck, Eva Cutiongco-de la Paz, Lay-Myint Yoshida, Stephane Hue, Maria Rosario Z. Capeding, Carmencita D. Padilla, R. Tedjo Sasmono, Julius Clemence R. Hafalla, Martin L. Hibberd.

**Investigation:** Ava Kristy Sy, James Ashall, Mary Anne Joy Reyes, Mary Ann Quinones, William Jones-Warner, Huynh Kim Mai, Le Thuy Lien, Hung Do Thai.

**Methodology:** Joseph R. Biggs, James Ashall, Oliver J. Brady, Joris Menten, Freya Rasschaert, Liesbeth Van Wesenbeeck, Julius Clemence R. Hafalla, Martin L. Hibberd.

**Supervision:** Ava Kristy Sy, Amadou O. Tandoc, Chihiro Iwasaki, Noriko Kitamura, Marnix Van Loock, Eva Cutiongco-de la Paz, Lay-Myint Yoshida, Stephane Hue, Maria Rosario Z. Capeding, Carmencita D. Padilla, R. Tedjo Sasmono, Julius Clemence R. Hafalla, Martin L. Hibberd.

**Visualization:** Joseph R. Biggs.

**Writing – original draft:** Joseph R. Biggs, Ava Kristy Sy, Oliver J. Brady, R. Tedjo Sasmono, Julius Clemence R. Hafalla, Martin L. Hibberd.

**Writing – review & editing:** Joseph R. Biggs, Ava Kristy Sy, James Ashall, Marsha S. Santoso, Oliver J. Brady, Mary Anne Joy Reyes, Mary Ann Quinones, William Jones-Warner, Amadou O. Tandoc, Nemia L. Sucaldito, Huynh Kim Mai, Le Thuy Lien, Hung Do Thai, Hien Anh Thi Nguyen, Dang Duc Anh, Chihiro Iwasaki, Noriko Kitamura, Marnix Van Loock, Guillermo Herrera-Taracena, Joris Menten, Freya Rasschaert, Liesbeth Van Wesenbeeck, Sri Masyeni, Sotianingsih Haryanto, Benediktus Yohan, Eva Cutiongco-de la Paz, Lay-Myint Yoshida, Stephane Hue, Maria Rosario Z. Capeding, Carmencita D. Padilla, R. Tedjo Sasmono, Julius Clemence R. Hafalla, Martin L. Hibberd.

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
