## [Decision Letter · Decision Letter 0]

1 Feb 2022

Dear Mr Biggs,

Thank you very much for submitting your manuscript "Combining rapid diagnostic tests to estimate primary and post-primary dengue immune status at the point of care" for consideration at PLOS Neglected Tropical Diseases. As with all papers reviewed by the journal, your manuscript was reviewed by members of the editorial board and by several independent reviewers. In light of the reviews (below this email), we would like to invite the resubmission of a significantly-revised version that takes into account the reviewers' comments. 

The reviwers expressed interests in a discussion or comparison of the method proposed in this study with other protocols published that have been used to classify primary and post-primary dengue infections, and how Zika infection would interfere with the analyzes

We cannot make any decision about publication until we have seen the revised manuscript and your response to the reviewers' comments. Your revised manuscript is also likely to be sent to reviewers for further evaluation.

Sincerely,

Ernesto T. A. Marques, M.D./Ph.D

Associate Editor

Rebecca Rico-Hesse

Deputy Editor

The reviwers expressed interests in a discussion or comparison of the method proposed in this study with other protocols published that have been used to classify primary and post-primary dengue infections, and how Zika infection would interfere with the analyzes

Reviewer's Responses to Questions

**Key Review Criteria Required for Acceptance?**

**Methods**

-Are the objectives of the study clearly articulated with a clear testable hypothesis stated?

-Is the study design appropriate to address the stated objectives?

-Is the population clearly described and appropriate for the hypothesis being tested?

-Is the sample size sufficient to ensure adequate power to address the hypothesis being tested?

-Were correct statistical analysis used to support conclusions?

-Are there concerns about ethical or regulatory requirements being met?

Reviewer #1: The study presented by Biggs et al. is well-conducted and included appropriate methods and analysis. However, this reviewers finds that there are some concerns to be addressed to improve the clarity and interpretability of their work.

1.a. Regarding the methodology, and although the study has been properly conducted, the first limitation that came to my attention was the choice of the standard serologic assay. Was any kind of accuracy comparison performed among the panbio test and other commercially available kits that would justify the choice of the first?

1.b. The authors state on page 11, lines 321-323 that participants’ DENV immune status classification was performed based on methods described in the reference #12. Although, unnecessary repetition of published data/methods shall be avoided, it is now clear to the reader how this classification was performed. Ref #12 describes the development of an algorithm to properly differentiate active from previous DENV infections based on demographic and serologic data and comparisons to WHO gold standard. In the current manuscript, it is not clear if the authors kept the same rationale or simply used the cut-off suggested by the Panbio test manufacturer.

Reviewer #2: Yes except where noted in the comments

Reviewer #3: -(261) Is there an EC approval from Vietnam?

-(263) How many verbals consent and assent were obtained (from 3 sites)?

-(298) N: 29,634 (28326+1217+200=29743)

-(310) NS1 RDT from Vietnam and Indonesia are different company?

-(313,317) SD Biosensor, Abbott?

-(323) How long IgM, IgG can be prolonged after infection?

-(333) Fever day 2 (PCR-, NS1-, IgM+, IgG+) = active post-primary or recent dengue infection?

**Results**

-Does the analysis presented match the analysis plan?

-Are the results clearly and completely presented?

-Are the figures (Tables, Images) of sufficient quality for clarity?

Reviewer #1: The conclusions presented by the current manuscript are in agreement with the generated data and performed analysis. However, information regarding cross-reactivity with other flavivirus is not discussed in the results/discussion section, although it is of utmost importance for the present work.

1.c. DENV active infection (as determined by PCR) and DENV serology (as determined by ELISA) seem to be well characterised in the study, however the authors have not included any data regarding cross-reactivity with other co-circulating flaviviruses. Indeed, the authors state on page 11, lines 335-336 that “Post-primary cases include current DENV infections with at least one previous flavivirus exposure.” As cases of flaviviruses co-infection and several hypothesis indicating more severe Dengue or Zika clinical phenotypes associated to previous flavivirus exposure, how would the authors explain the applicability of the proposed diagnostic framework? Also, how did the authors rule out the possibility of flavivirus co-infection and further DENV misclassification?

Reviewer #2: Yes

Reviewer #3: -(S2 Table) Why Indonesia showed equal DENV1-4 

-S3 Table result came from which country?

-(433) Indonesia and Vietnam results were similar or different?

-(445) Table 2 is the Vietnam result?

-(460) Different NS1 RDTs (from Vietnam) showed similar agreement?

-(470) Why didn’t compare with the actual RDT result?

-(471) Is there any change of being negative in Table 3

**Conclusions**

-Are the conclusions supported by the data presented?

-Are the limitations of analysis clearly described?

-Do the authors discuss how these data can be helpful to advance our understanding of the topic under study?

-Is public health relevance addressed?

Reviewer #1: The conclusions were appropriately drawn based on the authors' findings.

Reviewer #2: Yes

Reviewer #3: -(495) Diagnosis of acute dengue infection from suspected dengue patients need confirmed lab test (PCR+) before combination of RDTs results can give clear predictions of immune status.

-(520,528) Please also compare with both NS1 RDTs companies.

-(552) How many severe dengue in primary and post-primary?

**Editorial and Data Presentation Modifications?**

Reviewer #1: A few minor concerns are listed below to improve clarity:

2.a. Dengue testing:

- S1 Table should be replaced by S2 Table on page 11, line 318. Also, please clarify the meaning of each symbol used in the table, according to the main text.

- S2 Table: Although information on how DENV immune status of the participants was determined at the time of enrolment is described in the main text, this information should also be mentioned in the table.

2.b. Data analysis:

- Adding a figure describing the diagnostic algorithm (or rationale) used in the study would greatly benefit the manuscript.

Reviewer #2: (No Response)

Reviewer #3: (No Response)

**Summary and General Comments**

Reviewer #1: The goal of this manuscript was to determine and validate a framework to properly differentiate samples from primary and post-primary Dengue infections based on different combinations of points of care tests. The authors provided a well-conducted study with interesting results, however, the main limitations seem to be the choice of the standard serologic test and the lack of information regarding cross-reactivity with other flavivirus (e.g.; Zika). These two factors would be crucial to validate the authors' data and the proposed diagnostic framework. Further comments are presented in the previous sections of this review.

Reviewer #2: Well written and interesting article on using combinations of tests to diagnose primary, secondary (+), historic or uninfected patients with suspected dengue. I have a few questions that I think would improve the manuscript if the authors could address them (immediately below) and then some minor points for their consideration.

It wasn’t clear why optimising a statistical measure (Youden’s J index) is the best way to select thresholds, especially if those thresholds are going to lead to decisions being made on clinical care. A more thoughtful determination of the relative importance of sensitivity and specificity is called for.

The authors have done logistic regressions of numerical test results to overall classification. I would have expected them to do two further things to demonstrate the validity of the logistic regressions: (1) to fit non-linear (on the logit scale) relationships for instance using splines to assure themselves and the reader that a linear relationship holds and/or (2) to present empirical estimates of the proportions to show that the logistic curve adequately characterises the data. This could be done through Hosmer-Lemeshow plots, or simply by aggregating all the points with IgM panbio below 10, from 10 to 20, 20 to 30 etc (say) and showing the empirical proportion and confidence interval is similar to that from the logistic regression.

The combined framework leads to quite some uncertainty on the disease category (Table 3) for some combinations, such as disease day 1 and 2 with 1 positive RDT. If Table 3 is meant as a look up table, would there be no benefit to using the numeric values presented in Fig 2 to reduce the uncertainty? I thought it odd to coarsen the data at this early stage.

Minor points:

182: Presumably the cases ‘sought’ care rather than ‘sort’ed care.

244: Awkward phraseology. “For instance, patients with a post-primary dengue infection could be given greater priority for monitoring and IHT than patients with primary dengue.” Would be better

255: Careful: elsewhere this is given as ‘days of disease’ rather than of infection.

607: If ethics permits it, it would be preferable to make the data available online without needing the intercession of the corresponding author.

Figure 1: The whiskers on the bars are not defined. I assume they are 95%Cis. If so, several of the whiskers extend beyond the legal range (all six panels in panel B and both the panels in panel C [by the way, shouldn’t the panels be labelled a to p rather than a to c?]). Furthermore, some are seemingly deterministic (four bars in three of the panels in panels B and C). If the sample sizes are too small for the normal approximation to be valid then the authors should use a different way to construct confidence intervals.

Reviewer #3: -The paper is interesting but difficult to follow so please include study flow chart.

-What are the reasons for multiple sites without comparison?

PLOS authors have the option to publish the peer review history of their article (what does this mean?). If published, this will include your full peer review and any attached files.

Reviewer #1: No

Reviewer #2: No

Reviewer #3: No
---

## [Decision Letter · Decision Letter 1]

28 Mar 2022

Dear Mr Biggs,

We are pleased to inform you that your manuscript 'Combining rapid diagnostic tests to estimate primary and post-primary dengue immune status at the point of care' has been provisionally accepted for publication in PLOS Neglected Tropical Diseases.

Best regards,

Ernesto T. A. Marques, M.D./Ph.D

Associate Editor

Rebecca Rico-Hesse

Deputy Editor

Reviewer's Responses to Questions

**Key Review Criteria Required for Acceptance?**

**Methods**

-Are the objectives of the study clearly articulated with a clear testable hypothesis stated?

-Is the study design appropriate to address the stated objectives?

-Is the population clearly described and appropriate for the hypothesis being tested?

-Is the sample size sufficient to ensure adequate power to address the hypothesis being tested?

-Were correct statistical analysis used to support conclusions?

-Are there concerns about ethical or regulatory requirements being met?

Reviewer #1: The revised version of the manuscript by Biggs and colleagues has clarified the concerns previously raised by this reviewer. The choice of the standard serological test to be used and the criteria for DENV immune status classification are now properly presented in the final version of this manuscript.

Reviewer #3: The study presented by Biggs et al. is well-conducted and included appropriate methods and analysis. Moreover, this revised version addresses all reviewer comments well.

**Results**

-Does the analysis presented match the analysis plan?

-Are the results clearly and completely presented?

-Are the figures (Tables, Images) of sufficient quality for clarity?

Reviewer #1: The impact of ZIKV co-circulation and antibody cross-reactivity on the study data was addressed by the authors. The concerns raised by this reviewer were fully addressed.

Reviewer #3: The results presented by the revised manuscript are in agreement with the generated data and performed analysis.

**Conclusions**

-Are the conclusions supported by the data presented?

-Are the limitations of analysis clearly described?

-Do the authors discuss how these data can be helpful to advance our understanding of the topic under study?

-Is public health relevance addressed?

Reviewer #1: -

Reviewer #3: The conclusions were appropriately drawn based on the authors' findings.

**Editorial and Data Presentation Modifications?**

Reviewer #1: Minor concerns/editing previously raised were properly addressed to better present data tables and analysis.

Reviewer #3: Accept

**Summary and General Comments**

Reviewer #1: This reviewer considers that the revised version of the manuscript by Biggs et al. addresses all the concerns previously raised and, therefore, this reviewer is satisfied with the comments and answers provided.

Reviewer #3: The authors provided a well-conducted study with interesting results.

PLOS authors have the option to publish the peer review history of their article (what does this mean?). If published, this will include your full peer review and any attached files.

Reviewer #1: No

Reviewer #3: No

---

## [Editor Report · Acceptance letter]

20 Apr 2022

Dear Mr Biggs,

We are delighted to inform you that your manuscript, "Combining rapid diagnostic tests to estimate primary and post-primary dengue immune status at the point of care," has been formally accepted for publication in PLOS Neglected Tropical Diseases.

Best regards,

Shaden Kamhawi

co-Editor-in-Chief

Paul Brindley

co-Editor-in-Chief
